# Exploring the Nexus between Food Systems and the Global Syndemic among Children under Five Years of Age through the Complex Systems Approach

**DOI:** 10.3390/ijerph21070893

**Published:** 2024-07-09

**Authors:** Aline Martins de Carvalho, Leandro Martin Totaro Garcia, Bárbara Hatzlhoffer Lourenço, Eliseu Verly Junior, Antônio Augusto Ferreira Carioca, Michelle Cristine Medeiros Jacob, Sávio Marcelino Gomes, Flávia Mori Sarti

**Affiliations:** 1School of Public Health, University of Sao Paulo, Sao Paulo 01246-904, Brazil; barbaralourenco@usp.br; 2Centre for Public Health, Queen’s University Belfast, Belfast BT12 6BA, UK; l.Garcia@qub.ac.uk; 3Institute of Social Medicine, Rio de Janeiro State University, Rio de Janeiro 20550-013, Brazil; eliseujunior@gmail.com; 4Nutrition Course, Health Sciences Center, University of Fortaleza, Fortaleza 60811-905, Brazil; carioca@unifor.br; 5Biodiversity and Nutrition Laboratory, Federal University of Rio Grande do Norte, Natal 59078-970, Brazil; michelle.jacob@ufrn.br; 6Department of Nutrition, Health Sciences Center, Federal University of Paraiba, João Pessoa 58051-900, Brazil; savio.gomes@academico.ufpb.br; 7School of Arts, Sciences and Humanities, University of Sao Paulo, Sao Paulo 03828-000, Brazil; flamori@usp.br

**Keywords:** food system, global syndemic, climate change, overweight, malnutrition, children

## Abstract

The intricate relationship between food systems and health outcomes, known as the food-nutrition-health nexus, intersects with environmental concerns. However, there’s still a literature gap in evaluating food systems alongside the global syndemic using the complex systems theory, especially concerning vulnerable populations like children. This research aimed to design a system dynamics model to advance a theoretical understanding of the connections between food systems and the global syndemic, particularly focusing on their impacts on children under five years of age. The framework was developed through a literature review and authors’ insights into the relationships between the food, health, and environmental components of the global syndemic among children. The conceptual model presented 17 factors, with 26 connections and 6 feedback loops, categorized into the following 5 groups: environmental, economic, school-related, family-related, and child-related. It delineated and elucidated mechanisms among the components of the global syndemic encompassing being overweight, suffering from undernutrition, and climate change. The findings unveiled potential interactions within food systems and health outcomes. Furthermore, the model integrated elements of the socio-ecological model by incorporating an external layer representing the environment and its natural resources. Consequently, the development of public policies and interventions should encompass environmental considerations to effectively tackle the complex challenges posed by the global syndemic.

## 1. Introduction

The global syndemic encompasses complex synergistic interactions between multiple co-occurring epidemics that share common drivers [1]. The concept of the global syndemic is pivotal in understanding the current exacerbation of obesity, undernutrition, and climate change in diverse countries worldwide, considering their interconnectedness. The uneven distribution in the occurrence of these phenomena across the globe presents distinct implications for countries located in the global south and north, thereby requiring nuanced examinations of their impacts. For instance, the dual challenge of underweight and obese individuals has been escalating across developing nations, primarily due to the rise in obesity rates, while being underweight persists in South Asian and African countries [2].

Children under five years of age represent a vulnerable demographic group within the global syndemic debate, considering the coexistence of obesity and undernutrition that underscores the need for comprehensive intervention strategies. In 2022, obesity affected 16% of the global adult population, and approximately 37 million children under the age of five were considered overweight, according to the World Health Organization [3]. The challenge of the double burden of malnutrition is particularly pronounced among children living in low- and middle-income countries. Although the United Nations Children’s Fund (UNICEF) reported progress in reducing stunting among children under five years of age in 2022 [4], wasting has remained at high levels, and at the same time, the amount of overweight children has continued to increase over the years. Malnutrition can result in morbidity, mortality, and disability, along with impaired cognitive and physical development. Nearly half of all deaths in children under five years of age are due to undernutrition (stunting, wasting, and being underweight), which heightens their risk of dying from common infections, increases the frequency and severity of these infections, and delays recovery [4,5]. This means that malnutrition among children under five years of age is a major public health concern, especially in low- and middle-income countries.

Furthermore, the intersection between socioeconomic, demographic, and environmental factors influencing health outcomes in various countries has been showing the significant impacts of climate change on indicators connected to the global syndemic. The link between food systems and health outcomes through the food-nutrition-health nexus has been shown to connect with environmental concerns, since food production is responsible for approximately one-third of anthropogenic greenhouse gas emissions [6]. Additionally, environmental changes are linked to food production and health outcomes of populations, representing a continuous cycle [1].

The global food system represents the pivotal link that intertwines diverse aspects of health and environmental concerns, embodying a complex network that influences the global syndemic. Food systems are characterized by intricate structures encompassing a wide array of stakeholders, being marked by linear and nonlinear interactions, feedback loops, and dynamic shifts over time [7]. The holistic understanding and design of strategic interventions in food systems may help to tackle the root causes and interconnected challenges of the global health crisis [8].

Previous studies have explored various facets of childhood-related problems within the phenomenon of the global syndemic (e.g., [9]). However, there is a gap in the literature regarding the assessment of food systems in conjunction with components of the global syndemic through the lens of the complex systems theory. Thus, the objective of the present study was to bridge this gap by exploring the interactions and feedback loops between the characteristics of the food systems and the factors linked to the global syndemic. The focus was on their impacts on children under five years of age, as malnutrition in this age group is a major public health issue among low- and middle-income countries, and it is exacerbated by climate change events.

The research aimed at designing the framework of a system dynamics model that will allow advancing the theoretical foundations involving the connections between food systems and the global syndemic components, with a particular focus on their impacts on children under five years of age, in addition to comprehensively capturing the potential entry points for practical interventions designed to halt the adverse effects of the global syndemic on vulnerable populations.

## 2. Materials and Methods

The framework proposed to identify the mechanisms of the system dynamics within food systems linked to the global syndemic was developed through the following two-fold approach: (1) a rapid literature review using global syndemic descriptors and (2) the authors’ insights into the relationships between the food, health, and environmental components of the global syndemic involving children under five years of age (Figure 1).

The rapid literature review was conducted through a systematized standard search, which is recommended for guiding complex systems modeling [10]. The search took place between January and February of 2024 using the Scopus and Pubmed databases, and it was based on the records returned by the following combination of terms in the title, abstract, and keywords: overweight AND undernutrition AND “climate change” AND child*.

We included only papers that met the inclusion and exclusion criteria established for the literature search. The inclusion criteria compromised studies published in English since the inception of the databases, with their full-text versions available online. The exclusion criteria consisted of reviews and studies that did not mention children under five years of age. Following the removal of duplicates and studies not aligned with the research scope, four articles were identified in the review, and the key connections between the causes and consequences of the global syndemic were extracted (Table 1).

Based on the connections identified in the review, the initial draft model underwent refinement through a collaborative effort involving the research team, which comprised experts from diverse disciplines including nutrition, economics, biology, and complex systems. The expert panel process involved a two-hour online meeting where the draft model, outlining the causes and consequences of the global syndemic, was presented. This was followed by discussions on what should be added, removed, or revised based on the researchers’ expertise and the paper’s objectives. During the process, three connections identified in the review were excluded due to lack of relevance to the study, eighteen additional connections emerged from the team meeting, and four connections were improved to promote the straightforward identification of the cause-and-effect connections in the model (Appendix A). For example, the connection between inadequate family food practices and undernutrition found in the literature was modified to reflect the connection between adequate eating habits by the family and children. This change was made to emphasize that it is not the family’s consumption that directly affects a child’s undernutrition, but rather the child’s diet, which, in turn, influences their level of undernutrition.

The final iteration of the model categorized the factors into the following five distinct topics: environmental, economic, school-related, family-related, and child-related, and each of these was represented by a unique color code within the conceptual model. The causal connections between factors were represented by arrows, which could either have a positive (+) or a negative (−) sign, indicating, respectively, a positive or a negative relationship between the factors. Subsequently, balancing (B) and reinforcing (R) feedback loops were identified and included in the model. While balancing loops are mechanisms that stabilize a system and prevent it from drifting too far from its intended equilibrium, reinforcing loops lead to changes in the same direction, amplifying an initial change and moving the system away from equilibrium. The loops could be explored according to their potential for comprising tipping points in the mitigation or prevention of the global syndemic [11].

The proposed model was also integrated into the socio-ecological approach [12]. Placed at the core, an individual is surrounded by interpersonal, institutional, and political factors according to this approach. These factors at different levels interact to produce diverse outcomes for the individual.

## 3. Results

The conceptual model presented the components of food systems, namely, food production, food environment, and food consumption and their impacts on environment and health among children under five years of age. The model was grounded in 17 factors (Table 2), along with 26 connections and 6 feedback loops, according to the factors identified in the literature review and expert knowledge (Figure 2).

The model presents indirect connections between the climatic and health components of the global syndemic among children under five years of age, with the environmental impacts represented on the right side of the model (shown in green) and the health impacts on the left side of the model (shown in yellow). The lack of direct connections between the factors linked to climate change and overweight children and undernutrition shows that the causal pathways are moderated by intermediate determinants connecting the two components of the wider system.

Considering the environmental context (green), three factors, six connections, and two feedback loops were identified in the conceptual model. Environmental loop 1 comprises a balancing loop, showing that staple food production (e.g., soybean, corn, etc.) represents a significant contributor to deforestation in some countries, which, in turn, leads to climate change. However, climate change may reduce the production of staple foods in both quality and quantity. Simultaneously, livestock production influence in the loop encompasses a major driver of environmental impacts due to its contributions to deforestation and climate change through the emission of greenhouse gas from animals, exacerbating the negative environmental impact on staple food production. Environmental loop 2 is also a balancing loop and refers to the action of climate change in increasing extreme weather events, i.e., floods and droughts, which may negatively affect the productivity of both livestock and fruit and vegetable production.

The economic context (light pink) presents three main factors related to food production, comprising an important pathway between the environment and food consumption. The connections of the livestock, staple, and fruit and vegetable outputs to affordable food prices showed economic relationships connecting the food systems production and consumption components, indicating the role of scale economies in productivity and resulting in affordable food prices, which are important determinants of food choices for families and schools. Supply and demand mechanisms regulate food prices, production, and consumption, generating a balancing feedback mechanism. It is worth noting that there are other factors that potentially influence food prices at local and global levels, e.g., the global economy and wars, among others. The conceptual model focused on economic elements that may be influenced by public policies at the local level, thus disregarding other disruptive factors out of the scope of the local government’s strategies (i.e., external shocks).

The school-related (blue) and family-related (red) contexts included six factors and two feedback loops related to the family and school environments surrounding children. The demand loops illustrated the dynamics of the purchase and availability of foods influencing the quality of children’s dietary consumption practices. Hence, the conceptual model considered purchase as a proxy for consumption, considering that children under five years of age primarily consume meals provided by their families or schools, and they have limited input into food selection. The model incorporated diverse factors influencing food purchase and consumption directly linked to food demand, including public policies, family dietary practices, family education, and family income, among others; yet there were additional determinants of food demand that were out of the scope of the present study.

The health outcomes of being overweight and undernutrition in children are represented on the right side of the model (yellow), and they lead to child morbidity and mortality. These effects are not limited to this stage of life, as there may be consequences in subsequent stages of life, with implications for the economic, social, labor market, and healthcare systems. The relationships between climate change and children’s health are influenced by several stakeholders, permeating diverse stages of the food systems through food production and distribution through food consumption.

The socio-ecological approach to the food systems and health outcomes used in the conceptual model is illustrated in Figure 3. In the present study, children were placed in the center. Interactions were highlighted with various factors from different dimensions, including interpersonal elements (i.e., parent–child relationships), organizational influences (i.e., schools), institutional interactions within communal dynamics (i.e., economic relationships), and, finally, policy frameworks (i.e., government strategies). Additionally, the model integrated the outermost component representing the environmental layer, recognizing that natural resources (i.e., water, energy, air, soil, etc.) and the conservation of various species are fundamental for food production and health outcomes in a population in the short-, medium-, and long-term. The colors used in Figure 3 are the same as those used in Figure 2 according to each dimension represented.

## 4. Discussion

The conceptual model proposed in the present study established and qualified connections between diverse factors, including components of the global syndemic involving being overweight, undernutrition, and climate change, focusing on children under five years old. The results showed the potential mechanisms and interactions among the agents in the food systems and health outcomes, taking into account the dynamic nature of the phenomena. Additionally, the conceptual model integrated elements of the socio-ecological model by incorporating an external layer representing the environment with natural resources, which play pivotal roles in influencing outcomes in other layers.

The left side of the model represented environmental impacts from food production. Recent evidence has shown that different food types exert distinct impacts on the environment. Notably, livestock, particularly beef production, represents resource-intensive food production, accounting for approximately 40% of land use and contributing approximately 57% of the total global food production greenhouse gas emissions [23,24,25,26]. On the other hand, the production of plant-based food generally represents lower environmental burdens in comparison to animal-sourced food production. However, deforestation for staple crops and livestock exacerbates the carbon footprints associated with food systems in many parts of the world [27]. Food processing and transportation were excluded from the model due to their lower environmental impacts, as food production represents approximately 80% of the greenhouse gas emissions related to food supply [28].

The environmental impacts of food production also influence food availability at the population level in terms of quality and quantity, generating feedback loops. Elevated CO_2_ concentrations have been associated with reduced levels of iron, zinc, and protein in cereals and legumes [29]. Despite the theoretical potential for growth in food productivity due to CO_2_, the increasing frequency of extreme climate events presents a detrimental effect on food production, particularly in developing countries with limited capacities for addressing climate change [30]. Additionally, the effects of extreme climate events are not limited to the local scale but can spread through an entire food system, affecting the supply chain [31]. For example, a drought in 2008 impacted rice and wheat production worldwide and, combined with stock market and political factors, resulted in an estimated 75 million people facing malnutrition and forcing approximately 130 million people into poverty [32].

Government interventions represent the main strategies for influencing food supply and demand, i.e., strategies in public policies for fostering healthy diets may propose links between taxes and subsidies on the production and consumption of certain foods to promote changes in the dietary patterns of a population. Yet, there may be unintentional consequences to government strategies, e.g., bandwagon or snob effects, generating changes in food systems towards an increase in demand due to aspirational consumption. There is evidence in the literature of policies that may support consumers. Systematic reviews have shown that taxes and subsidies can effectively reduce the consumption of sugar-sweetened beverages and increase fruit and vegetable purchases [33,34,35]. According to the reviewed studies, providing subsidies for healthy foods, ranging from 1.8% to 50%, led to a significant increase in their consumption. The magnitude of this increase was at least half the size of the tax applied [34]. On the other hand, sugar-sweetened beverage taxes, ranging from 5% to 30%, resulted in decreases in their consumption ranging from 5% to 48%, with an impact on consumption proportional to the tax applied. The Healthy Incentive Pilot Study was a clinical trial that investigated the impact of a 30% rebate on fruit and vegetable purchases for the Supplemental Nutrition Assistance Program (SNAP). They found increases of 23% and 30% in fruit and vegetable purchases, respectively [36].

The food environments in which families are embedded have repercussions on children’s dietary intake and health. Family educational level, for instance, influences healthy eating behaviors in children and adolescents. It has been shown that children of mothers with high educational levels have more chances to eat fruits and vegetables whereas children of mothers with low educational levels consume more sugar, fat, and protein [37]. Family nutrition has been affected by changes in global dietary patterns, leading to the substitution of traditional food habits, based on minimally processed foods, with the consumption of ultra-processed products and homogeneous diets with low diversity. In the United Kingdom, for instance, 68% of the energy consumed by children comes from ultra-processed products [38]. The speed at which these changes occur differs between modern (also referred to as conventional) and traditional food systems. In modern food systems, the scientific evidence suggests that, for instance, proximity to fast-food outlets and ultra-processed products is directly related to increased childhood obesity [39]. Conversely, in traditional food systems, children are more exposed to the double burden of malnutrition, with a high prevalence of undernutrition, but there has been growth in the incidence levels of overweight children and obesity [40]. Thus, the food environment shapes the lifestyle, cultural norms, and parental eating habits of families, consequently impacting the routines and eating habits of children [41].

School environments are also directly related to children’s eating habits, serving as an institutional pathway for cultural transmission regarding food. Although there is a possible lack of access to pre-primary school in low- and middle-income countries, the Sustainable Development Goals (SDG), explicitly, target 4.2, aim to “ensure that all girls and boys have access to quality early childhood development, care, and pre-primary education so that they are ready for primary education”. Among the 18 studies included in a systematic review, 17 showed positive results regarding the impacts of interventions in the school environment on improving children’s diets and reducing their weights [42]. Investing in institutional purchases from family agriculture for schools has also been a strategy used to increase the availability and access to healthy foods in schools. School feeding programs benefit approximately 388 million children worldwide, and governments increasingly recognize their multiple benefits for populations, such as social protection and food security for students. When linked to family agriculture, these programs can contribute to the development of shorter supply chains closer to schools, along with the provision of local and culturally appropriate foods [43]. Nonetheless, the primary hurdles stem from inadequate investment in family agriculture and inefficient logistics linking the field to the school, potentially compromising the quality of school meals. Feasible remedies encompass initiatives fostering investment in agricultural policies and the coordination of family farmers [44].

The conceptual model proposed in the present study was designed to allow for the identification of potential strategies within public policies fostering healthy lifestyles and sustainable food system actions to tackle the effects of the global syndemic among children. It is important to emphasize that climate change, being overweight, and undernutrition comprise multifactorial phenomena influenced by numerous elements. However, climate change represents a trigger for acute changes in food systems, e.g., droughts and floods, which cause decreases in the productivity of certain food items, resulting in price hikes due to product shortages in markets. Although there is a wide variety of foods available worldwide, including the possibility of substitution among food items to maintain diet quality, vulnerable individuals have limited access to food, and thus, they are susceptible to food insecurity and famine due to extreme climate events. The model further illustrated that prioritizing sustainable food systems is essential for emphasizing environmental conservation and improving the well-being of families and children.

The study presents certain limitations. First, we did not fully account for the diverse conditions that can vary across countries and contexts. Instead, we developed a general framework, primarily focusing on low- and middle-income countries. We acknowledge that this is an initial exploration into the complex interplay between food systems and the global syndemic among children under five years of age, and it attempts to connect different fields. Although it is an important step forward in the scientific understanding of these interactions, it requires validation by food system stakeholders. Yet, the proposal was grounded in a systematized literature review, and it was validated through experts’ opinions. Second, the conceptual model focused on food, health, and environmental elements within the reach of strategies in public policies designed by local governments, i.e., it excluded external shocks (economic crises and wars, among others) and other influential mechanisms connected to the global syndemic, i.e., physical activity and infectious diseases. However, it is important to acknowledge that model comprised simplifications for a representation of the reality to allow for the identification of key factors linked to the phenomena under investigation, and thus, such a model must strike a balance between complexity and utility for the purposes of its analysis.

Finally, the conceptual model presented in the study represents the first step in the development of more complex system approaches that allow for the measurement of outcomes resulting from the interconnections included within their conceptual frameworks. First, this conceptual model will be refined, tested, and validated in the next stages of the research. After that, we will develop agent-based models and system dynamics models that allow for the investigation of outcomes resulting from the interconnections included within the conceptual framework. These models will help identify strategies within public policies tackling deforestation while ensuring food accessibility, quality, diversity, and affordability, alongside alterations in household and school environments for reducing the prevalence of overweight children and undernutrition. Ultimately, these efforts aim to understand how a food system can be more sustainable and resilient to future challenges.

## 5. Conclusions

The exploration of the dynamic interactions between food systems and global syndemic factors among children portrayed in the conceptual model showed the complexity and interconnectedness of the system, in addition to the potential mechanisms for designing strategies in public policies to foster healthy and sustainable diets for children within the environment-food-health nexus. The examination of environmental, economic, school-related, and family-related feedback loops provided valuable insights to the identification of entry points that allow for mitigation of the impacts of the global syndemic on children. Additionally, the integration of the conceptual model with the socio-ecological model included the boundary of the outermost component, representing the environmental layer, highlighting the importance of the conservation of natural resources for the survival of the species. Therefore, the design of public policies and interventions should consider environmental factors to effectively address the complex challenges posed by the global syndemic, especially among vulnerable population groups like children.

## Figures and Tables

**Figure 1 ijerph-21-00893-f001:**
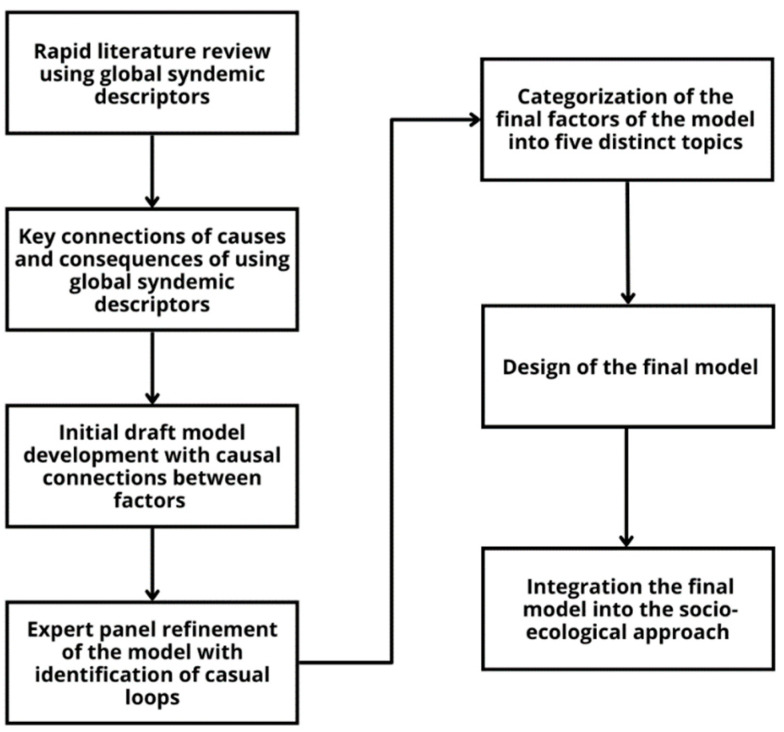
The approach to the framework development.

**Figure 2 ijerph-21-00893-f002:**
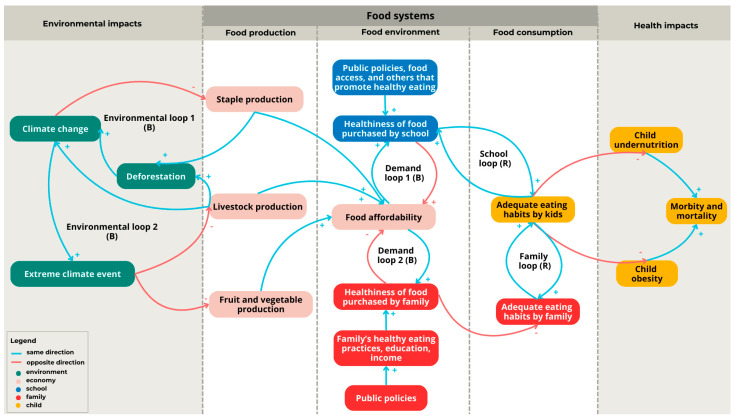
Conceptual model of the environment-food-health nexus among children under five years of age using the complex systems approach: (B) the balancing feedback loop, (R) the reinforcing feedback loop, (+) positive relationship between the factors, (−) negative relationship between the factors.

**Figure 3 ijerph-21-00893-f003:**
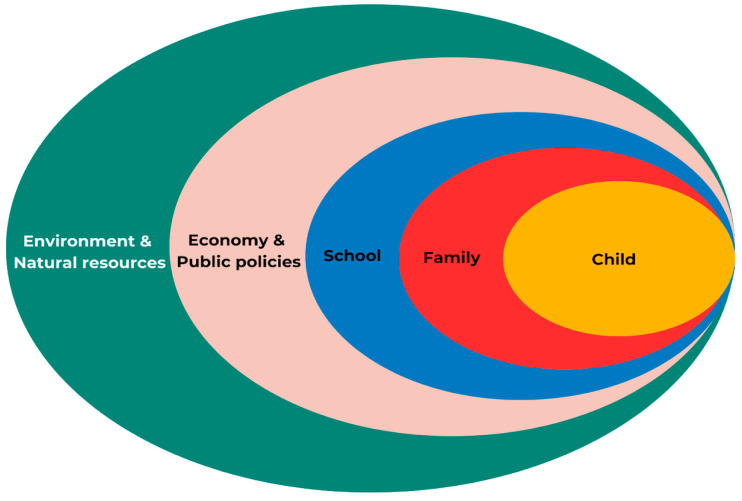
Dimensions of the conceptual model of the environment-food-health nexus among children.

**Table 1 ijerph-21-00893-t001:** Key connections between the causes and consequences of the global syndemic identified in the rapid literature review.

Cause		Consequence	Reference
Low physical activity	-->	**Being overweight**	[11]
Low family income	-->	**Undernutrition**	[12,13]
Inadequate food consumption	-->	**Being overweight and suffering from undernutrition**	[11,12]
Lack of social protection services and poor health access	-->	**Being overweight and suffering from undernutrition**	[14]
Inadequate family food practices	-->	**Being overweight and suffering from undernutrition**	[11]
Low educational levels in parents	-->	**Being overweight and suffering from undernutrition**	[11]
**Being overweight and suffering from undernutrition**	-->	Mortality	[13]
**Being overweight and suffering from undernutrition**	-->	Morbidity	[11,12,13,14]
Food production (animal production, high impact; plant production, low impact)	-->	**Climate change**	[13]
**Climate change**	-->	Food system	[13]
**Climate change**	-->	Food-borne and other infectious diseases	[11]

Bold text: global syndemic indicators.

**Table 2 ijerph-21-00893-t002:** Conceptual model factors and their definitions.

Group	Factor	Definition	Reference
Children	Adequate eating habits of the kids	Patterns of food consumption and dietary behaviors exhibited by children that support their growth and health	[15]
	Child undernutrition	Health conditions resulting from inadequate food intake or insufficient nutrient intake, which can lead to stunting, being underweight, or wasting	[16]
	Children being overweight	Being overweight is a condition of excessive fat deposits, and it is measured by a child’s weight-for-height that is greater than two standard deviations above the child growth standard median	[3]
	Morbidity and mortality	Prevalence of illness or death within a population	[17]
Family	Public policies	Presence of public policies that can influence family behaviors and characteristics	
	A family’s healthy eating practices, education, and income	Factors related to a family including the dietary habits adopted by a family, their levels of education, and the income within a family	
	Healthiness of food purchased by a family	Healthiness of food purchased to provide meals for consumption within a household	
	Adequate eating habits by a family	Patterns of food consumption and dietary behaviors exhibited by families that support their well-being and health	[15]
School	Healthiness of food purchased by a school	The food items purchased to provide meals for students	
	Public policies, food access, and others that promote healthy eating	Factors related to a school including its menu guidelines, public policies that can influence its food purchases, and the availability of food	
Economy	Livestock production	Breeding, raising, and management of animals such as cattle	[18]
	Staple production	Cultivation, harvesting, and processing of food crops that form the primary dietary components for a population, such as corn and soybean	[19]
	Food affordability	Ability of families to purchase and access an adequate quantity and variety of food	[20]
	Fruit and vegetable production	Cultivation, harvesting, and processing of fresh fruits and vegetables	[21]
Environment	Climate change	Anthropogenic alterations in the earth’s climate patterns, including changes in temperature, precipitation, wind patterns, and other environmental indicators	[22]
	Extreme climate event	A weather phenomenon that deviates from the average during a specific time period, including unusually high or low temperatures, droughts, floods, hurricanes, and other weather extremes	[22]
	Deforestation	Removal of trees and forests from an area, typically for the purpose of clearing land for agriculture	[22]

## Data Availability

No new data were created in this study. Data sharing is not applicable to this article.

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
