# Peer review of "Exploring the Nexus between Food Systems and the Global Syndemic among Children under Five Years of Age through the Complex Systems Approach"

_ijerph, 2024, doi:10.3390/ijerph21070893_

Round 1

Reviewer 1 Report

Comments and Suggestions for Authors

Dear Editors,

The Manuscript prepared explores and presents the influence of factors involved in food systems and the global syndemic on health outcomes in children under five years of age.

The text is well prepared and adequately analyses the interactions between socioeconomic, demographic and environmental factors in health outcomes, manifested mainly by malnutrition and obesity in children.

The methodology used was a brief literature review and an expert panel process involving the research team. Based on the main connections of causes and consequences between the two main systems. A theoretical model proposed presents five distinct topics - environmental, economic, school-related, family-related and child-related.

The conceptual model presents 16 factors including the components of the global syndemic, along with connections and six feedback loops, according to the factors identified in the literature review and experts' knowledge.

The tables are ok.

Figure 1 is difficult to understand; it requires the reader to pay more sophisticated and in-depth attention to understand the direction of the intermediate components that link the causal pathways of a complex system. I have doubts if it helps, as the text is better prepared.

In my view, factors related to family's educational level could be better explored beyond the role of school meals. Excellent formal education promotes a more enlightened and critical generation of people to make relevant and appropriate food choices.

I congratulate the authors for their work and reflection.

Reviewer 2 Report

Comments and Suggestions for Authors

Manuscript titled “Exploring the Nexus Between Food Systems and the Global Syndemic Among Children Under Five Years of Age through the Complex Systems Approach” reports an analysis between different variables (family, public policy, schools, etc.) that contribute to the health or disease of children. The topic is interesting, although the analysis performed is, in this reviewer’s opinion, not critical or justified enough. Please see specific comments for details:

1.       The manuscript focuses on children up to five years old. The various problems mentioned like obesity and undernutrition can still affect older children and teenagers, in fact, they may be masked at such young age and only become apparent after this age. Please consider providing a clearer justification in the introduction, about why you decided to place this cutoff at five years of age.

2.       Related to the previous comment, not all children attend school before five years old, or only do so for a brief enough time that it may not be enough to exert a significant effect during this period. Thus, this factor may not be relevant for a significant number of children.

3.       The number of articles considered is too low to support the analysis or conclusions reached. The fact that the authors focused on young children may have contributed to excluding many studies, however, this should be an indicator of the need to restructure the analysis and/or literature search performed in order have sufficient support for your review.

4.       The population of the studies and various other data should be provided, since the environmental, societal, economic, and various other conditions of children in different countries (and even in different places within the same country) are likely to be vastly different, therefore making it difficult to group them together.

5.       Table 1 contains “Key connections of causes and consequences…”. These appear to be results, thus, would it be possible to move the table to that section?

6.       In Table 2, why did you group “Family's healthy eating practices, education, income, food access, public policies, and others” into a single factor? Education and income could be independent factors that have an impact on a family’s dietary choices. Moreover, a family does not have control over “food access” and “public policies”.

7.       Also in Table 2, “Food purchase by school” and “Public policies, food access, and others” may be the same factor. For example, what schools buy and offer to children is dictated by public policy, or is it assumed that schools independently decide what to buy, prepare and offer to children?

Reviewer 3 Report

Comments and Suggestions for Authors

This great article attempts to solve the food-health nexus in preschool children. The methods and elaboration of the model are quite sound. I do have some remarks:

- Figure 1: water is not present in this scheme (or is this included in nutrition?).

- An additional figure could be added, containing a scheme with the strategy to get to this model.

Results: a sixth dimension could be added to the model: energy.

Discussion: a sustainable food system has been chosen, but the disadvantages of a conventional food system could be mentioned.

Reviewer 4 Report

Comments and Suggestions for Authors

This manuscript aims to design a system dynamics model to advance theoretical understanding of the connections between food systems and the global syndemic, particularly focusing on their impact on children under five years of age. There are similar studies in this area, one of which is the paper: https://doi.org/10.3390/nu16081119 and this publication should be a complementary reference manuscript to this. 

Other comments:

L73. AIM of the study must also be addressed from the perspective of describing the contribution to the field under analysis and the elements of scientific novelty presented, as the LAST, SEPARATE paragraph of this section, to be easier visible. Develop it better. What differentiate your paper from others on the same topic? Give a reason for interest in this paper.

The quality of the figures must be improved!

Before the conclusion section, please provide a section that discusses "future implications and direction"

How is the "Experts’ panel decisions on the connections between causes and consequences of the global syndemic" carried out? What method or approach do you use?

- Authors also mention the risks of obesity and undernutrition, but do not explain in more depth regarding undernutrition in the Stunting aspect, it is necessary to provide a more comprehensive picture in the current manuscript.

- Also discuss the sustainability of Food Systems.

Round 2

Reviewer 2 Report

Comments and Suggestions for Authors

Manuscript titled “Exploring the Nexus Between Food Systems and the Global Syndemic Among Children Under Five Years of Age through the Complex Systems Approach” reports an analysis between variables that contribute to the health or disease of children. The present version of the document was revised according to comments and suggestions made during an initial revision.

After reading the modified version and the authors’ reply, it is apparent that they considered and addressed every comment and suggestion made by the present reviewer. However, the main flaw of the work still persists, since the conclusion reached by the authors is, in this reviewer’s opinion, not properly supported by the analysis made and the number of references analyzed. The topic remains quite relevant, but it may be too ambitious to properly support with the information analyzed since, as the authors themselves mention, “We acknowledge that the number of studies included was limited”. Perhaps it may be more appropriate to perform such a complex analysis on local/regional variables and population, since those would be more homogenous and easier to reach a more substantiated conclusion. Or maybe limiting the scope could be another approach, instead to trying to tackle such an ample topic.

Author Response

Dear Reviewer,

Thank you for your feedback.

As the title of our work says, it represents an initial exploration into the complex interplay between food systems and the global syndemic among children under five years of age. We understand that this is an important step forward in the scientific understanding of these interactions, though it is not conclusive. Our intention was to contribute to the body of knowledge in this area with insights, recognizing that science advances incrementally through the accumulation of ideas and evidence.

Regarding your specific concerns about the support for our conclusions, it is important to highlight that the limited number of studies included in the review reflects the gap in the literature on the subject. The exploration of an innovative subject (i.e., nexus between food systems and global syndemic among children using complex systems approach) should also be considered in the assessment of the findings in the literature review. Thus, the literature search recovered a number of studies that is representative of the recent advances in the area. Nevertheless, we believe that the revised literature, combined with the authors' expertise, provides valuable insights that can guide future research efforts in the field.

We appreciate both your suggestions to focus on local or regional variables and populations, and to potentially narrow the scope of the study to enhance the depth of analysis. These are excellent considerations to expand upon the knowledge established by this initial exploratory article, and they are part of our future plans.

To address your concern, we included some phrases in the discussion section.